# Settlement of a Foundation on an Unsaturated Sandy Base Taking Vibrocreep into Account

Armen Z. Ter-Martirosyan *, Alexander N. Shebunyaev and Evgeny S. Sobolev

Department of Soil Mechanics and Geotechnical Engineering, National Research Moscow State Civil Engineering University, 26, Yaroslavskoye Shosse, 129337 Moscow, Russia; shebunyaev95@mail.ru (A.N.S.); soboleves@mgsu.ru (E.S.S.)
* Correspondence: gic-mgsu@mail.ru

**Abstract:** Dynamic loading causes (1) a substantial change in the strength and deformation parameters of sandy soil and (2) excessive viscoplastic deformation. The goal of this study is to create a new analytical solution to the problem of the settlement of (1) the foundation that is the source of dynamic loading, and (2) a nearby foundation, taking into account the rheological properties of sandy soil subjected to vibration, given that these rheological properties depend on shear stresses. The proposed solution allows the progress of deformation over time to be described. The present paper states and provides an analytical solution for the problem of evaluating the settlement of a single foundation that transmits static and dynamic harmonic pressure to the base. The authors also analyze the settlement of another foundation located at some distance from the transmitting foundation. The second foundation transmits static pressure to the base. The dependence of the viscosity coefficient on the shear stress intensity and vibration intensity, as well as the vibrocreep decay over time, are based on the exponential and homographic dependencies previously identified by two of the authors (A.Z. Ter-Martirosyan and E.S. Sobolev). The solution to the problem is obtained by numerical integration in the Mathcad program of an analytical expression for nonlinear viscoplastic deformations. As a result of the research, the authors have found that the dynamic viscoplastic component makes the greatest contribution to foundation settlement. The settlement of the transmitting foundation increases along with increasing static and dynamic pressure transmitted to the base. The settlement of the nearby foundation increases when the pressure increases under the foundation, but it reduces when static pressure from the transmitting foundation, the depth of the foundation, and the distance between the foundations increase. General analytical dependencies obtained by the authors comply with the results of laboratory and field experiments performed by other researchers. These dependencies can be used to predict the settlement of foundations in whose unsaturated sandy bases mechanical vibrations propagate.

**Keywords:** excessive foundation settlement; foundation vibrations; dynamic cyclic loading; vibrocreep of sandy soil; viscosity of sandy soil; numerical integration in Mathcad; mathematical analysis of nonlinear soil deformation

**MSC:** 74L10

## 1. Introduction

The safety of industrial and civil buildings and structures must be evaluated throughout their lifecycles if these buildings and structures are subjected to dynamic effects from various sources [1,2]. Such sources encompass various fixed and mobile machines and mechanisms, traveling road and rail vehicles, earthquakes, etc. In practice, dynamic loading may have a wide range of adverse effects on the performance of bases and foundations, ranging from slow long-term accumulation of settlements to higher-than-normal continuous settlements, causing critical damage to foundations and superstructures.

For example, according to R.A. Ershov and A.A. Romanov, the average annual settlement of buildings located within 30 m from the axis of highways is 0.3–2.2 mm/year [3]. Any excessive settlements, caused by the cyclic loading of the sand base, can also worsen the operating parameters of the equipment installed inside due to warping and eccentricities [4]. Cases of excessive settlement, reaching 28 cm, were registered in the plant building area where a swaging machine was in operation. As for another plant building, the settlements of column foundations reached 40 cm in the area where a drop forging hammer was in operation [3]. According to the observations of D.D. Barkan [5,6], a drop forging hammer weighing 4.5 tons that was in operation in a production shop caused excessive settlement of an adjacent brick building located within the plant territory. Eventually, its settlement led to its failure. In his work, V.M. Pyatetsky [7] provides an extensive list of 25 facilities where critical structural damage was observed (cracks in bearing walls, excessive deformations of the framework, deformations of crane tracks, disintegrated column–truss joints, etc.). Settlements were observed at a distance of up to 20–30 m from the source of dynamic loading (compressors, sawmills, crushing machines, ore grinders, etc.). According to other researchers, the maximum value of settlement reaches 88 cm [3,8–10].

Changes in the deformation and strength properties of sandy soils subjected to dynamic loads are important factors in predicting the stress–strain state of the bases of building structures [1,2,11]. Engineering practice is aware of a large number of structural failures caused by the vibrocreep of sandy soils. These failures were manifested as the substantial settlement of bases (including those described above), damaged building structures, and the collapse of entire buildings, which most clearly demonstrate the relevance of this issue. A study on the phenomenon of vibrocreep of sandy soils was initiated in the first half of the 20th century and it gained increasing interest in 1960–1980 due to the widespread implementation of industrial construction projects accompanied by the installation of powerful sources of dynamic loading [12]. Most of the works in this subject area focus on the study of the deformation and stability of saturated sandy bases since pore pressure increases under dynamic loading and vibration-induced liquefaction of the foundation occurs, which can lead to the rapid destruction of structures. However, many authors study the deformation of unsaturated sandy soils.

An important contribution to the study of vibrocreep processes was made by the outstanding scientist D.D. Barkan [1,2], who conducted initial experiments focused on the identification of the viscosity coefficient. In the course of his experiments, a ball was immersed in sandy soil under vibration. D.D. Barkan conducted a simple laboratory experiment to demonstrate that sandy soil develops rheological properties if subjected to vibration, and the viscosity coefficient of sandy soil depends on vibration acceleration. Of particular interest, is the work of V.A. Ilyichev, V.I. Kerchman, B.I. Rubin, and V.M. Pyatetsky, in which the authors described a large-scale field experiment consisting of the field observation of vibrations and settlement of seven experimental foundations of different sizes and pressure under the foundation. In this experiment, one of the foundations was the source of the vibrations [7,10,13]. The authors discovered the effect of static and dynamic pressure under the foundation on the manifestation of vibrocreep. V.S. Bogolyubchik, M.N. Goldstein, and V.Ya. Khain conducted numerous flute and field tests to simulate foundation displacement under dynamic loading [14–16]. Results obtained by D.F. Gil et al. from flume testing [17] demonstrate a decrease in the stability of the sandy base under cyclic loading when there is an increase in the frequency and amplitude of vertical vibrations. Flume tests, involving the cyclic loading of a sandy base model, were conducted by K. Al-kaream et al. [18] and showed the accumulation of plastic deformations over time with the final stabilization of the deformation process. The work of J. Wang et al. [19] reports the results of large-scale laboratory studies on the mutual effect of two closely located foundations on a sandy base subjected to cyclic loading based on the depth and distance between them. The authors show that if the two foundations are subjected to cyclic loading, the greatest bearing capacity is observed if the distance between the foundations is equal to twice the width of the foundation base. The works

report the regularity of settlement development over time at different values of soil density, static loading, and foundation depth. The conclusions drawn by the above researchers are important practical materials for evaluating the results of the theoretical solution to the problem described in this work.

Early studies on the behavior of soils under dynamic loading were conducted by A. Casagrande and W.L. Shannon during the construction of the Panama Canal [20]. The tremendous loss of life and destruction of civil infrastructure facilities built on saturated sand bases and the collapses of hydraulic facilities as a result of the earthquakes in the US and Japan in 1960–1970, coupled with the launched construction of nuclear power plants, served as the reason for the intensive research of the behavior of sandy soils under dynamic loading. The publication written by H.B. Seed and K.L. Lee [21] reaches an important conclusion about the effect of lateral compressive pressure $\sigma_3$ on the number of loading cycles that cause soil failure. Studies by K.L. Lee and J. Focht [22] showed that, for practical purposes, the relationship between dynamic strength and effective pressure $\sigma_3$ can be taken as a direct dependence for a small range of stresses. Researchers R.C. Chaney and H.Y. Fang [23] studied the behavior of dry sandy soils subjected to cyclic loading and showed that axial deformation accumulates in the course of small-amplitude dynamic loading, and, following a number of cycles, the hysteresis loop closes and the specimen stabilizes. According to V.K. Khosla and R.D. Singh [24], if the dynamic stress amplitude $\sigma_d$ increases or the lateral compressive pressure $\sigma_3$ decreases, the stable state of the soil is replaced by the slow accumulation of deformations (vibrocreep).

A wide range of modern studies on the vibrocreep of sandy soils was conducted by two of the current study's authors (AZT-M and ESS) and Z.G. Ter-Martirosyan. The authors believe that under compression an increase in plastic deformations decays alongside an increase in the number of cycles and that in the case of low-frequency cyclic loading (up to 1–2 Hz), frequency has no significant effect on deformation propagation (the difference does not exceed 2% for the same number of loading cycles), while the number of loading cycles and soil density have a strong effect when the number and amplitude of the cyclic loading are at a maximum [25,26]. Vibrocreep coefficients for compression and shear differ significantly (up to 10 times). Hence, soils are more sensitive to shear vibration than to compression vibration [25,27]. In cases of cyclic and vibratory loading, additional deformations increase with increasing static shear stresses and decrease with increasing mean stresses. Therefore, these deformations depend on the proximity to the limiting state [25,27,28]. In the case of a vibratory action, vibrocreep-induced deformation increases with increasing frequency and decays in time [25,27]. Viscosity increases during loading, and by the time the vertical deformations stabilize, viscosity values approach the values obtained during the tests conducted at low velocities (nearly static ones). Viscosity plummets at the onset of loading, which causes vertical deformations to spike. Then, as the testing progresses, soil density increases, viscosity gradually increases, and vertical deformations stabilize. At the onset of a dynamic test, the viscosity differs from the "static" viscosity by a factor of 100. Hence, during dynamic loading, soil behaves as a non-Newtonian fluid [25,27,29]. E.A. Voznesensky conducted a wide range of research on the behavior of sandy bases and evaluated the development of deformations over time and the effect of the amplitude of dynamic pressure [12].

Liu X. et al. [30] compare the deformation of sand specimens subjected to static and cyclic loading and demonstrate the effect of cyclic loading and relative density on cyclic strength. M. Poblete et al. [31] describe a succession of sand tests whereby sand was subjected to various values of compressive pressure and vibration amplitudes under cyclic loading. B. Kafle and F. Wuttke [32] have found that in the case of cyclic loading, dry sand accumulates deformations more intensively than wet unsaturated sand. Each cycle makes the accrual of plastic strain smaller, but even with a large number of cycles, the strain does not reach any final value.

K. Wang et al. [33] studied the effect of lateral compressive pressure on the intensity of axial and volumetric deformation of sandy soil subjected to cyclic loading. It was

recorded that at a certain value of lateral compressive pressure the specimen did not collapse due to significant expansion (dilatancy), as it was recorded at lower values of $\sigma_3$, but its compaction occurred due to the increasing axial deformation and destruction of soil particles, which was also mentioned in the work of one of the current authors (AZT-M) [25]. This paper also describes the breakdown of deformation mechanisms of sandy soil subjected to cyclic load into plastic compaction, vibrocreep, and continuous plastic failure. The authors describe the deformation of sandy soil using an analytical dependence in which values of vertical $\sigma_1$ and lateral $\sigma_3$ stresses are used. Similar results are reported in the studies of Y. Jianhong et al. [34], where the authors use the case of triaxial tests to demonstrate three stages of soil deformation before failure: the elastoplastic stage, the slow creep stage, and the fast creep stage at different values of compressive pressure and vertical loading. The work by Dong-Ning D. et al. [35] has an analytical description of the rheological model describing this staged viscoplastic behavior of sandy soils subjected to vibration. The results of research performed by Y. Liu et al. [36] show a significant increase in transverse deformation in the case of cyclic loading; this feature is observed up to a certain value of axial deformation (in the authors' work it is 6%), and if it is exceeded, cyclic loading has no effect on transverse deformation even at different values of relative density and amplitude of cyclic deformation. The article written by D.F. Gil et al. [17] reports the effect of the frequency of dynamic loading on soil deformation.

Z. Wang and L. Zhang [37] show the effect of cyclic deformation amplitude, lateral compressive loading, the ratio of vertical and lateral static stresses, and vibration frequency on axial deformation and stiffness of sandy soil subjected to cyclic loading. A. Kumar et al. [38] present the results of triaxial tests of embankment soil of a railroad track at different vibration frequencies and show that an increase in frequency reduces soil stiffness to some extent, but any further increase in frequency does not lead to a substantial reduction in stiffness. The results of D. Song et al. [39] and W. Ma et al. [40] show how the shear modulus changes under cyclic loading at different angular deformation and compressive pressure values. It is found that the deformation modulus decreases significantly with increasing angular deformation during vibration, but this decrease reduces significantly with increasing compressive pressure. In addition, W. Ma et al. [40] investigated the effect of the intensity of static and dynamic components of loading on a decrease in its shear modulus. The authors provide an analytical relationship to take into account a decrease in the shear modulus using experimental parameters and the value of angular deformation. P. Xia et al. [41] propose an analytical dependence describing the vibration of soil under dynamic loading and breakdown into elastic and plastic components. Dependencies identified by the authors in respect of strain and shear moduli are also applicable to this study in terms of a reduction in the viscosity coefficient if the soil deformation over time and damping creep are considered.

The numerous works mentioned above offer a nearly exhaustive description of the deformation of sandy soil over time under cyclic loading taking into account the effect of the frequency and amplitude of dynamic impacts, the static loading, and the lateral pressure. This study, describing the process of deformation of sandy soil under dynamic loading, employs a rheological model that takes into account dependencies obtained earlier by the researchers of NRU MGSU of the effect of vibration acceleration, the proximity to the limit state, and the nature of deformation over time [25–28]. These dependencies are consistent with the results obtained by other researchers.

Publicly available articles offer numerical solutions to the problem of settling foundations resting on sandy non-saturated bases and subjected to dynamic loading (e.g., works written by Vivek P. and Ghosh P. [42], Wichtmann T. et al. [43], Wuttke F., Schmidt H.G., Zabel V. et al. [44]) or analytical solutions obtained by introducing reduced deformation characteristics of soils into soil analysis (e.g., Yi F. [45,46] and Pradel D. [47] suggest calculating settlement using angular deformation by taking into account a reduction in the shear modulus and soil porosity; one of the current authors (ESS) [48] suggests reducing the deformation modulus of the base by taking into account the vibrocreep coefficient).

At the same time, there are no analytical solutions obtained using the rheological model of sandy soil that allow for the accurate identification of dependencies and mathematical analysis of the effect of each factor separately. To fill this gap, the authors propose a new analytical solution to the problem of settlement of a single foundation subjected to dynamic harmonic loading and an adjacent foundation subjected to static loading.

Hence, the researchers identified several dependencies determining the deformation of sandy non-saturated soil for various parameters of dynamic loading. However, a review of solutions to problems of foundation settlement shows that authors do not use any currently available rheological models to describe deformation in their computational models. Moreover, the heterogeneous stress–strain state is formed in a foundation with varying proximity of tangential stress intensity to its limit value, which should also be taken into account in projections of foundation settlements. The purpose of this study is to propose a new solution to the analytical problem of foundation settlement under dynamic loading taking into account the vibrocreep of sandy soil with a changing viscosity coefficient, which depends on the stress level, vibration intensity, and time.

## 2. Materials and Methods

We consider the propagation of the settlement of a single foundation that has width $b_1 = 2 \cdot a_1$, and is located at depth $h_1$ (see Figure 1). The foundation is subjected to static load $N_1$ and dynamic harmonic load $\Delta N \cdot sin(\omega \cdot t)$, which trigger pressures $p_1$ and $\Delta p \cdot sin(\omega \cdot t)$ below the foundation base, resting on the homogeneous unsaturated sandy base, characterized by gravity $\gamma$, internal friction angle $\varphi$, deformation modulus $E$, Poisson's ratio $\nu$ and viscosity coefficient $\eta_0$ during manifestations of vibrocreep, the rheological parameter of strengthening at vibration creep $\alpha$ and coefficient $\delta$, showing the dependence of viscosity on vibration acceleration using the case of a refractory tube mill (width of foundation $b_1 = 7.2$ m, static pressure below the foundation base $p_1 = 160$ kPa, the amplitude of dynamic pressure $\Delta p = 30$ kPa, frequency of loading $\omega = 10$ Hz).

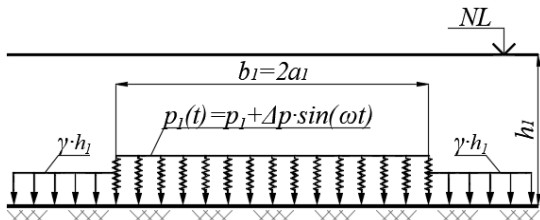

**Figure 1.** Loading diagram.

The base has a layer of unsaturated fine sand having the following principal design characteristics: $\gamma = 19.6$ kN/m$^3$; $\varphi = 30.3°$; $E = 30.2$ MPa; $\nu = 0.3$; $\alpha = 0.015$; $\Delta = 9.9$; $\eta_0 = 2.76 \cdot 10^5$ Pa·s. Information about the particle size distribution is provided in Figure 2.

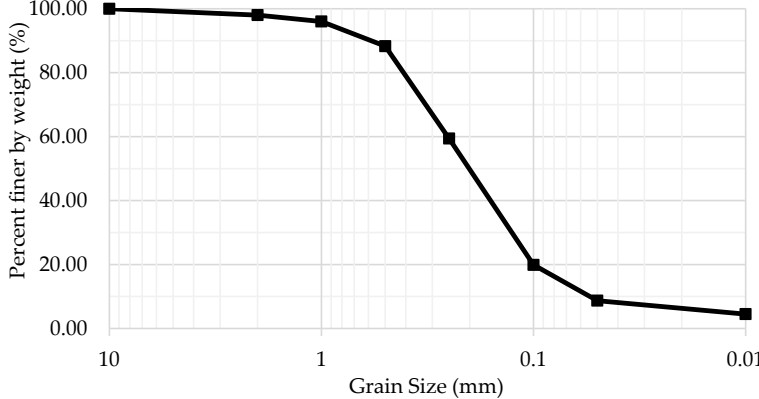

**Figure 2.** Particle size distribution in a specimen of sand.

A solution is provided for the case of the quasi-dynamic problem statement (with inertial terms being neglected in equations of motion) [49]: the sand medium becomes viscous due to the vibration triggered by the dynamic component of loading [3,6,8,50], and the foundation sinks into the viscous base due to the static component of vertical loading [51]. The foundation settlement in time $s(t)$ is composed of an elastic component $s_{st}$, which the foundation gets from the static component of loading $p_1$ before the commencement of operation, and a viscoplastic component $s_{dyn}(t)$, which the foundation gets after the commencement of equipment operation that has a dynamic effect on the base $\Delta p \cdot sin(\omega \cdot t)$ (1) [3].

$$s(t) = s_{st} + s_{dyn}(t) \tag{1}$$

Elastic settlement $s_{st}$ в is calculated according to the well-known dependence (2) [52].

$$s_{st} = \sum_{i=1}^{n} 0.8 \cdot \frac{\sigma_{zp,i} \cdot h_i}{E_i} \tag{2}$$

where $n$ is the number of layers within the compressible thickness;

$\sigma_{zp,i}$ is an additional stress in the $i$-th soil layer according to Formula (5);

$h_i$ is the thickness of the $i$-th soil layer;

$E_i$ is the static deformation modulus of the $i$-th soil layer.

The viscoplastic component of settlement $s_{dyn}(t)$ is calculated by Formula (3) by integrating the vertical deformations $\varepsilon_z$ over the depth of the sand layer [53] using the dependences obtained by authors AZT-M [25,28] and ESS [27] in respect of the propagation of vibrocreep in time, the effect of increasing shear stresses that approach their limit value, and the effect of vibratory acceleration on the viscosity coefficient.

$$s_{dyn}(t) = \int \varepsilon_z^{dyn} dz = \int_0^H \left[ \frac{\tau_i}{\alpha \cdot \eta_0 \cdot e^{-\Delta \cdot a \cdot \frac{\tau_i^* - \tau_i}{\tau_i^*}}} \cdot \left(1 - e^{-\alpha \cdot t}\right) \right] dz \tag{3}$$

where $\tau_i$ is the intensity of shear stresses according to Formula (4);

$\tau_i^*$ is the ultimate tangential stress according to Formula (4);

$\alpha$ is the experimental rheological hardening parameter [25,27];

$\Delta$ is an experimental coefficient, showing the dependence of viscosity on vibratory acceleration of vibrations [25,27];

$a$ is the vibratory acceleration of vibrations;

$\eta_0$ is the viscosity coefficient before the commencement of vibratory loading;

$H$ is the distance from the foundation to the bottom of the sand layer.

$$\tau_i^* = \sigma_m \cdot tg\varphi$$
$$\sigma_m = \frac{\sigma_x + \sigma_y + \sigma_z}{3} \tag{4}$$
$$\tau_i = \frac{\sqrt{\left(\sigma_x - \sigma_y\right)^2 + \left(\sigma_y - \sigma_z\right)^2 + \left(\sigma_z - \sigma_x\right)^2 + 6 \cdot \left(\tau_{xy}^2 + \tau_{yz}^2 + \tau_{zx}^2\right)}}{\sqrt{6}}$$

In the case of the application of a uniformly distributed load, expressions for stresses are obtained by G.V. Kolosov (5) [52,54] by integrating the solution of the Boussinesque problem in the case of two dimensions in width $2a$ for a plane strain problem [53,55].

$$\sigma_x = \frac{p}{\pi} \cdot \left(arctg\frac{a-x}{z} + arctg\frac{a+x}{z}\right) - \frac{2ap}{\pi} \cdot \frac{z \cdot \left(x^2 - z^2 - a^2\right)}{\left(x^2 + z^2 - a^2\right)^2 + 4a^2z^2}$$
$$\sigma_z = \frac{p}{\pi} \cdot \left(arctg\frac{a-x}{z} + arctg\frac{a+x}{z}\right) + \frac{2ap}{\pi} \cdot \frac{z \cdot \left(x^2 - z^2 - a^2\right)}{\left(x^2 + z^2 - a^2\right)^2 + 4a^2z^2} \tag{5}$$
$$\sigma_y = v \cdot (\sigma_x + \sigma_z) = \frac{2vp}{\pi} \cdot \left(arctg\frac{a-x}{z} + arctg\frac{a+x}{z}\right)$$
$$\tau_{xz} = \frac{4ap}{\pi} \cdot \frac{x \cdot z^2}{\left(x^2 + z^2 - a^2\right)^2 + 4a^2z^2}$$

By substituting expressions for stress components (5) into expressions (4) at $a_1$ = 3.6 m and $p_1$ = 160 kPa in Mathcad software, we obtain graphs for mean stresses $\sigma_m$ and shear stress intensity $\tau_i$, shown in Figures 3 and 4, respectively.

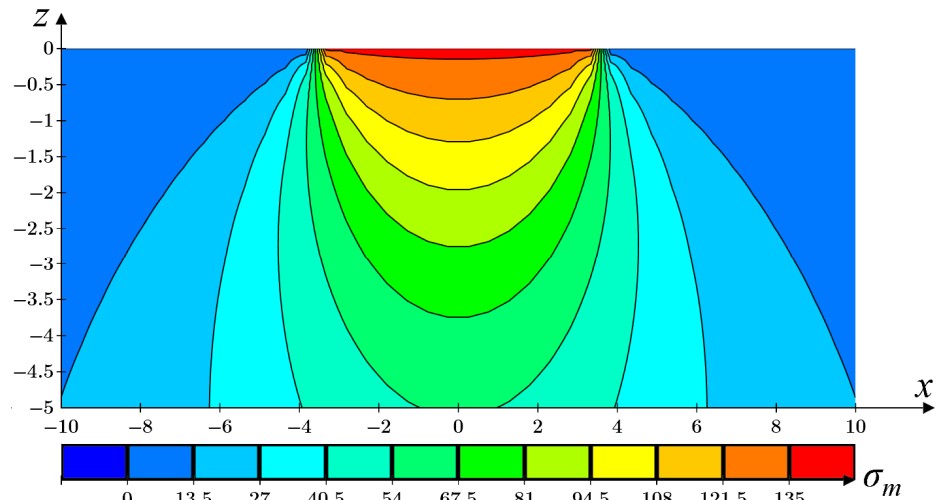

**Figure 3.** Isofields of mean stresses $\sigma_m$ in the sandy base at $a_1$ = 3.6 m and $p_1$ = 160 kPa.

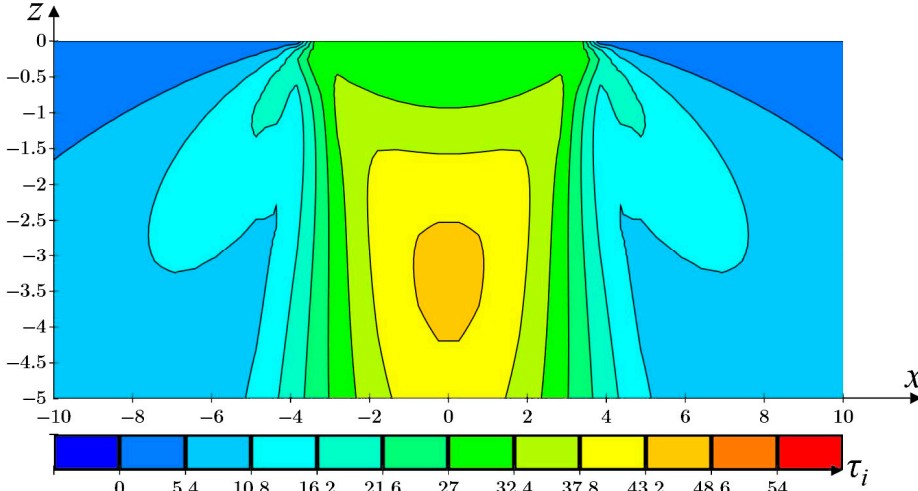

**Figure 4.** Isofields of the intensity of shear stresses $\tau_i$ in the sandy base at $a_1$ = 3.6 m and $p_1$ = 160 kPa.

The value of the vibration amplitude $A_{z,0}$ is found using the well-known Formula (6) [3].

$$A_{z,0} = \frac{\Delta N}{K_Z} \cdot \frac{1}{\sqrt{\left(1 - \frac{\omega^2}{\lambda_z^2}\right)^2 + (\Phi_z \cdot \omega)^2}} \tag{6}$$

where $K_Z$ is the stiffness coefficient of the base;

$\lambda_z$ is the frequency of natural vibrations;

$\Phi_z$ is the modulus of decay.

The base stiffness coefficient $K_Z$ is calculated by Formula (7) [3,9,10,52]. The expression for the coefficient $C_z$ in Formula (7) was adopted based on the results of a study by S.K. Lapin, who conducted field experiments on more than 300 foundations with an area of 0.5 to 3700 m² [3]. These results have satisfactory convergence with field experiments and are accepted in the Soviet standard of building design.

$$K_Z = C_z \cdot F = b \cdot E \cdot \left(1 + \sqrt{F_0/F}\right) \cdot F \tag{7}$$

where $b$ is the coefficient, taken as being equal to 1.0 for sands;
$E$ is the soil modulus of elasticity;
$F_0 = 10$ m$^2$;
$F$ is the area of the foundation bottom.
The frequency of the natural vibrations $\lambda_z$ is calculated by Formula (8) [3,7,10].

$$\lambda_z = \sqrt{\frac{K_z}{M}} \tag{8}$$

The approximate value of the modulus of decay $\Phi_z$ can be taken from reference tabular data depending on the type of soil [13] or found by Formula (9) [3].

$$\Phi_z = \frac{2\xi_z}{\lambda_z} = \frac{4\sqrt{M}}{\sqrt{p_m \cdot K_z}} \tag{9}$$

where $M$ is the mass of the oscillating object;
$p_m$ is the average pressure under the foundation.
The amplitude of the vertical vibrations of the soil $A_z(r)$ at distance $r$ from the center of the base of the source of vibrations is determined by Formula (10) [7,9,10,52].

$$A_z(r) = A_{z,0} \cdot \left\{ \frac{1}{\Delta \cdot \left[ 1 + (\Delta - 1)^2 \right]} + \frac{\Delta^2 - 1}{\sqrt{3\Delta}(\Delta^2 + 1)} \right\} \tag{10}$$

where $A_{z,0}$ is the amplitude of vertical vibrations of the foundation that is the source of vibrations;
$\Delta = r/r_0$ is the parameter that is equal to the ratio of distance $r$ from the center of the base of the foundation that is the source of vibrations to the reduced radius of that base.
The distribution of vibrations in depth $z$ follows dependence (11) [3,7,52]. It is noteworthy that the amplitude of harmonic vibrations with constant frequency is directly proportional to the acceleration of vibrations, as in a second derivative of displacement, so the following dependence (11) is also true for the distribution of vibration amplitudes with depth.

$$w = w_0 \cdot e^{-\beta z} \tag{11}$$

where $w$ is the acceleration of vibrations in depth $z$;
$w_0$ is the acceleration of the base vibrations at the level of the foundation bottom;
$\beta$ is the coefficient of decay, whose value is 0.07–0.10 m$^{-1}$ for sandy soils.
The expression for the vibration amplitudes of the base $A_z$ in Mathcad is shown in Figure 5. Vibration amplitude isofields for the sandy base $A_z$ obtained in Mathcad, are shown in Figure 6 for the case where the dynamic load component $\Delta p = 40$ kPa and $\omega = 10$ Hz.

$$A_z\left(x, z, \Delta p, p_1\right) := \text{if } |x| \leq r_0$$
$$\left\| A_{z0}\left(\Delta p, p_1\right) \cdot e^{\beta \cdot z} \right.$$
$$\text{else}$$
$$\left\| A_{z0}\left(\Delta p, p_1\right) \cdot \left( \frac{1}{\left| \frac{x}{r_0} \right| \cdot \left( 1 + \left( \left| \frac{x}{r_0} \right| - 1 \right)^2 \right)} + \frac{\left( \frac{x}{r_0} \right)^2 - 1}{\sqrt{3 \cdot \left| \frac{x}{r_0} \right| \cdot \left( \left( \frac{x}{r_0} \right)^2 + 1 \right)}} \right) \cdot e^{\beta \cdot z} \right.$$

**Figure 5.** Expression for vibration amplitudes of the base $A_z$ in Mathcad.

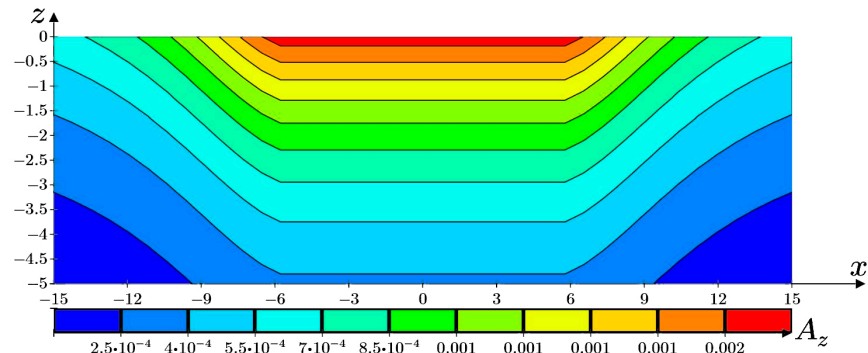

**Figure 6.** Isofields of vibration amplitudes of the base $A_z$ at $\Delta p$ = 40 kPa and $\omega$ = 10 Hz.

The expression for the viscoplastic component of settlement $s_{dyn}$ in Mathcad is shown in Figure 7. The expression is obtained by substituting expressions (7–9) in (6), and then (6), (10), and (11) in (3) [56,57].

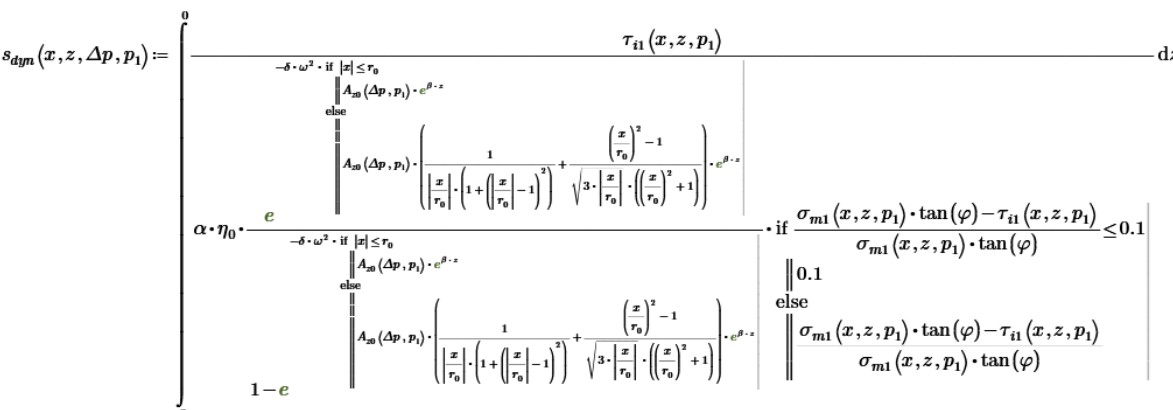

**Figure 7.** Expression for the viscoplastic component of settlement $s_{dyn}$ in Mathcad.

The above problem of settlement of a single foundation helps to solve another problem of excessive settlement of the foundation of a nearby building. Towards this end, we use the case of an existing finished product screening shop near which a tube mill will be erected. Figure 8 shows photographs of the site that will accommodate the tube mill.

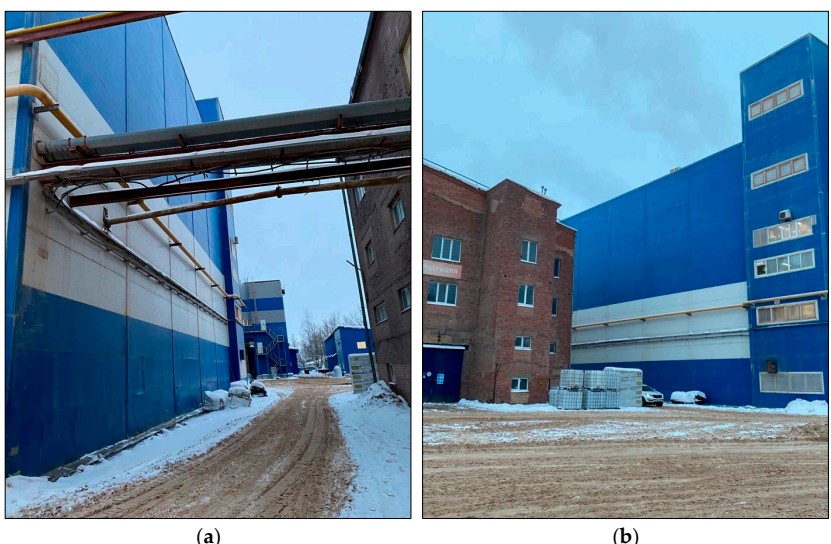

(**a**)　　　　　　　　　　　　(**b**)

**Figure 8.** (**a**,**b**) The tube mill site between existing workshops.

The foundation plan is shown in Figure 9 and the loading diagram is shown in Figure 10.

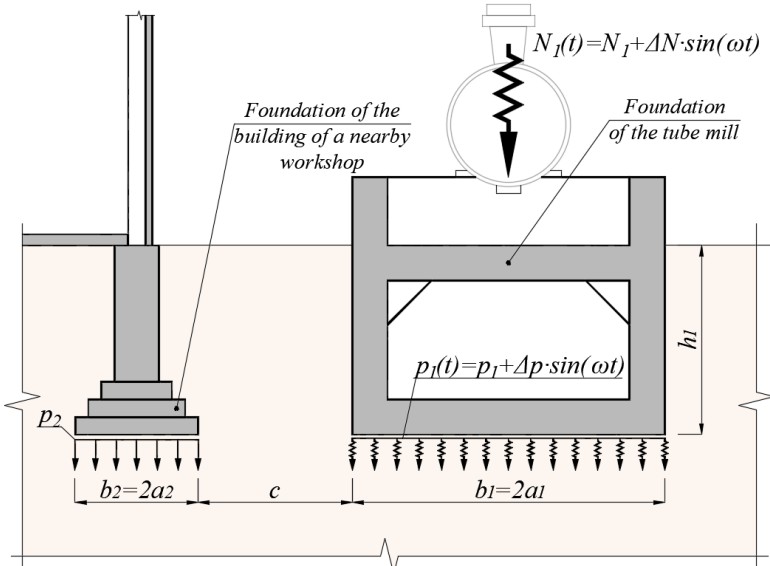

**Figure 9.** The foundation base plan of the tube mill, which is the source of dynamic loading, and the foundation plan of the building of a nearby workshop.

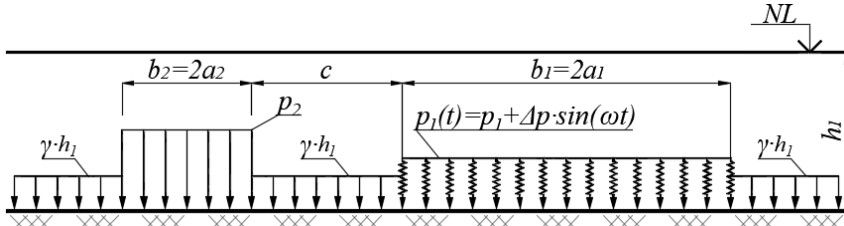

**Figure 10.** Loading diagram.

By placing the source of loading close to the existing foundations, we trigger additional stresses identified using the expressions obtained by G. Kolosov (5) [52,54].

Figures 11 and 12 show the isofields of mean stresses $\sigma_m$ and the intensity of shear stresses $\tau_i$ in the base subjected to the combined action of loads from the constructed foundation $p_1 = 160$ kPa and the existing foundation $p_2 = 350$ kPa if the half-widths of foundations are $a_1 = 3.6$ m, $a_2 = 1.2$ m, the distance between foundations $c = 4.0$ m, and their depth is $h_1 = 3.8$ m.

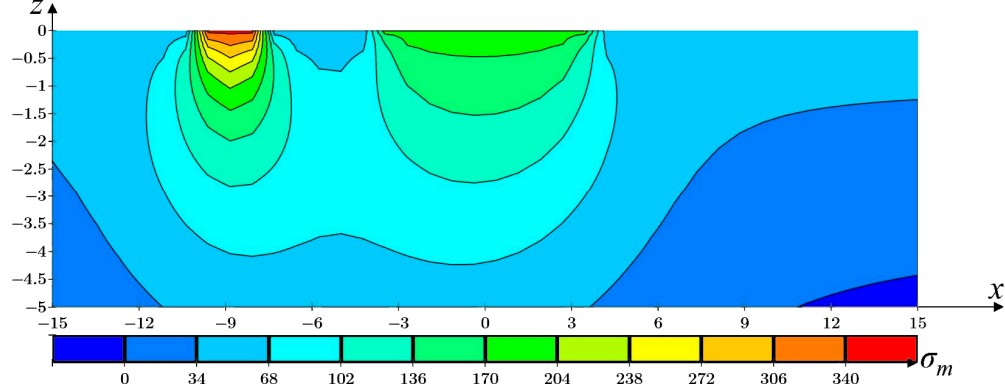

**Figure 11.** Isofields of mean stresses $\sigma_m$ in a sandy base.

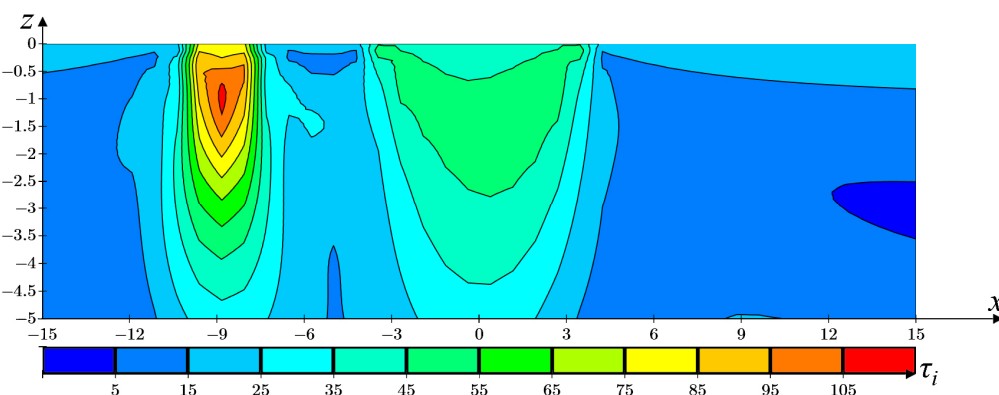

**Figure 12.** Isofields of stress intensity $\tau_i$ in a sandy base.

## 3. Results

As for a single foundation (Figure 1), if we sum the elastic $s_{st}$ and viscoplastic $s_{dyn}(t)$ components of the settlement calculated using Formulas (2) and (3), we will have a family of foundation settlement curves, as shown in Figure 13.

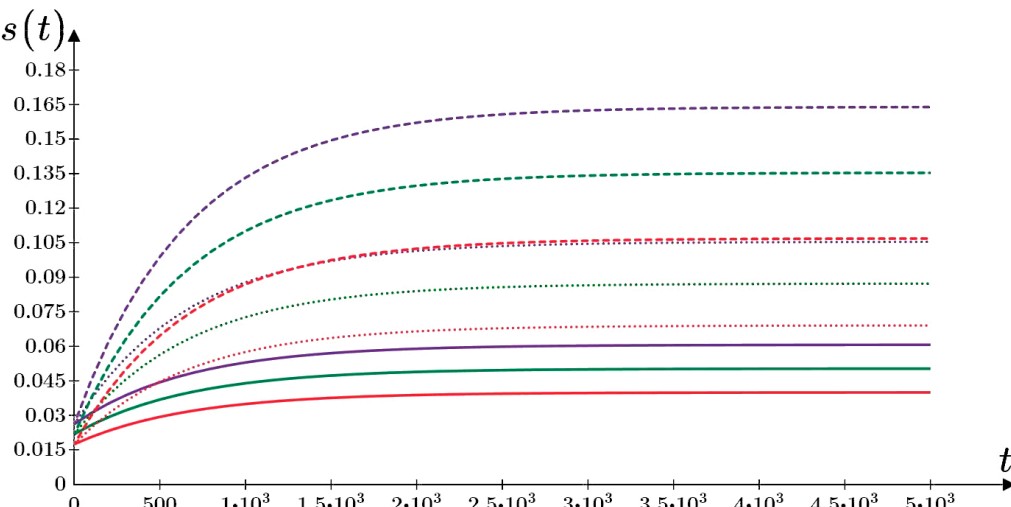

**Figure 13.** The propagation of foundation settlement in time at static loading $p_1$ and dynamic loading $\Delta p$: (————)—$p_1$ = 160 kPa, $\Delta p$ = 10 kPa; (⋯⋯⋯⋯)—$p_1$ = 160 kPa, $\Delta p$ = 20 kPa; (----------)—$p_1$ = 160 kPa, $\Delta p$ = 30 kPa; (————)—$p_1$ = 200 kPa, $\Delta p$ = 10 kPa; (⋯⋯⋯⋯)—$p_1$ = 200 kPa, $\Delta p$ = 20 kPa; (·----------·)—$p_1$ = 200 kPa, $\Delta p$ = 30 kPa; (————)—$p_1$ = 240 kPa, $\Delta p$ = 10 kPa; (⋯⋯⋯⋯)—$p_1$ = 240 kPa, $\Delta p$ = 20 kPa; (----------)—$p_1$ = 240 kPa, $\Delta p$ = 30 kPa.

Analysis of Figure 13 makes it clear that, as the static load component $p_1$ and dynamic load component $\Delta p$ increase, the foundation settlement increases. Hence, according to the results of Kerchman's field studies [58], an increase in the dynamic component $\Delta p$ makes a more substantial contribution by triggering viscoplastic shear of the double-sided extrusion type. This is evident in the nonlinear dependence $s(\Delta p)$ presented in Figure 14. The increase in settlement that accompanies an increase in dynamic component $\Delta p$ is caused by a decrease in the viscosity coefficient and an increase in vibratory acceleration of the base vibration, while the increase in settlement that accompanies an increase in static component $p_1$ is triggered by an increase in the intensity of shear stresses in base $\tau_i$ and their approaching ultimate stresses $\tau_i{}^*$, according to authors AZT-M [25,28] and ESS [27] and others [17,26]. This conclusion, stemming from the theoretical solution, complies with the results of large-scale field tests conducted by V.A. Ilyichev, V.I. Kerchman, B.I. Rubin, and V.M. Pyatetsky [7,13] focused on the influence of static and dynamic loading.

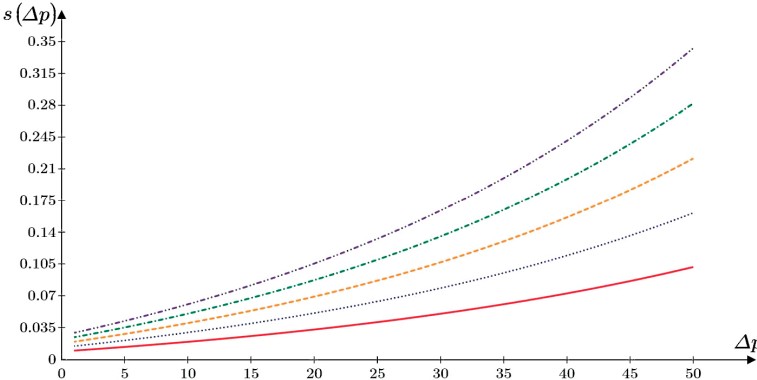

**Figure 14.** Dependence of the value of excessive foundation settlement on the value of the dynamic component of loading $\Delta p$ at different values of the static component of loading $p_1$: (——)—$p_1$ = 80 kPa; (··········)—$p_1$ = 120 kPa; (– – – –)—$p_1$ = 160 kPa; (–·–·–·–)—$p_1$ = 200 kPa; (··–··–··)—$p_1$ = 240 kPa.

If we take the values of the amplitude of vibrations and patterns of their propagation in plan and at depth from Formulas (6), (10), and (11), and analyze the propagation of viscoplastic settlement taking vibrocreep into account under dynamic loading according to Formula (3), we obtain the graph of settlement for a nearby foundation shown in Figure 15.

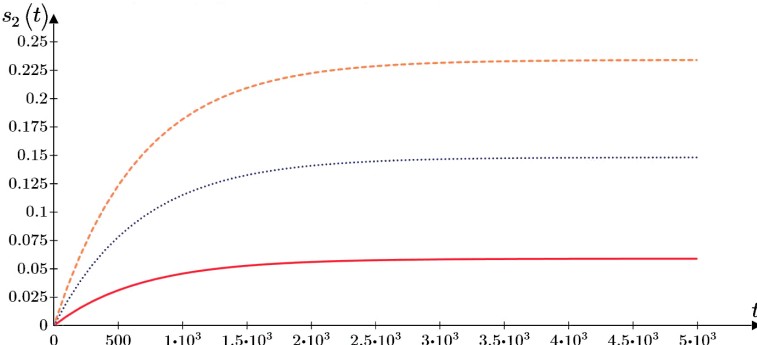

**Figure 15.** Foundation settlement propagation in time, if the value of static load $p_1$ = 160 kPa and dynamic load $\Delta p$ = 30 kPa, triggered by the source foundation, when static load $p_2$ from the foundation of a nearby building is: (——)—$p_2$ = 160 kPa; (··········)—$p_2$ = 240 kPa; (– – – –)—$p_2$ = 320 kPa.

If the dynamic component $\Delta p$ = 30 kPa and values of load $p_2$ are variable, the dependence of the value of excessive settlement of the foundation of a nearby building $s_2$ on the static load from the source foundation $p_1$ is as shown in Figure 16.

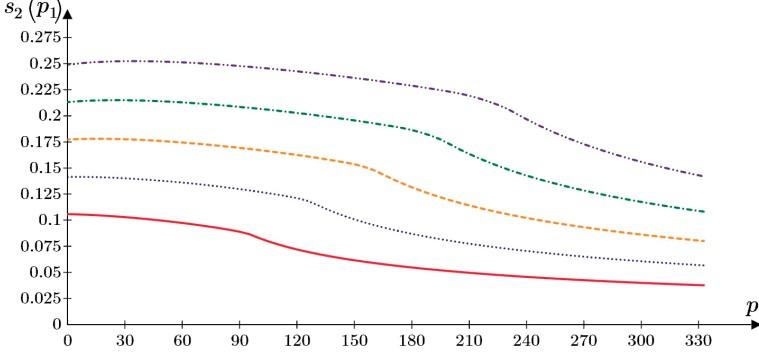

**Figure 16.** Dependence of the value of excessive settlement of the foundation of a nearby building $s_2$ on the static load from the source foundation $p_1$ if the dynamic component $\Delta p$ = 30 kPa, the distance between foundations $c$ = 4.0 m, and the load values $p_2$: (——)—$p_2$ = 160 kPa; (··········)—$p_2$ = 200 kPa; (– – – –)—$p_2$ = 240 kPa; (–·–·–·–)—$p_2$ = 280 kPa; (··–··–··)—$p_2$ = 320 kPa.

## 4. Discussion

When analyzing the $s_2(p_1)$ graph (Figure 16), one can observe a decrease in the settlement of a nearby foundation with an increase in the static load from the source foundation $p_1$, which can be explained by an increase in the difference between the intensity of shear stresses $\tau_i$ (Figure 17b) and their limiting value $\tau_i^*$, which in turn depends on the value of mean stress $\sigma_m$ (Figure 17a).

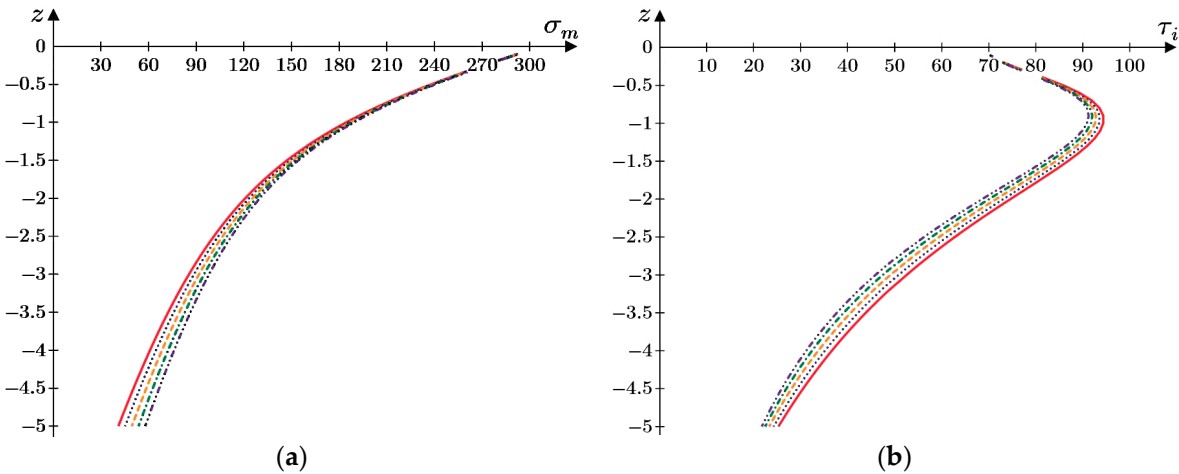

(a)          (b)

**Figure 17.** Dependence of stresses $\sigma_m(z)$ (**a**) and $\tau_i(z)$ (**b**) arising below the nearby foundation if loading $p_2 = 300$ kPa and loading values $p_1$: (——)—$p_1 = 160$ kPa; (··········)—$p_1 = 200$ kPa; (‑‑‑‑‑)—$p_1 = 240$ kPa; (‑·‑·‑·‑)—$p_1 = 280$ kPa; (··‑··‑··)—$p_1 = 320$ kPa.

It will be more obvious to show the effect of the load from the source foundation $p_1$ on the settlement of the nearby foundation using the coefficient $k_{ult}$ (12). This coefficient is part of Formula (3) and shows a decrease in the viscosity coefficient $\eta_0$ with the approach of the intensity of shear stresses $\tau_i$ to their limiting value $\tau_i^*$. Figure 18 clearly shows an increase in the coefficient $k_{ult}$ under the nearby foundation with an increase in the pressure $p_1$ of the source foundation.

$$k_{ult} = \frac{\tau_i^* - \tau_i}{\tau_i^*} \tag{12}$$

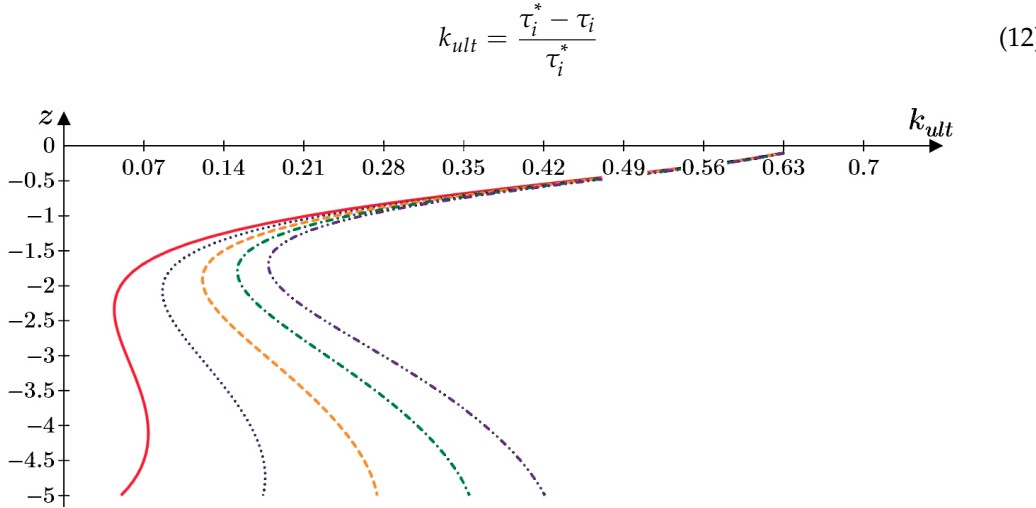

**Figure 18.** Dependence of the coefficient $k_{ult}(z)$ below the nearby foundation if loading $p_2 = 300$ kPa and loading values $p_1$: (——)—$p_2 = 160$ kPa; (··········)—$p_2 = 200$ kPa; (‑‑‑‑‑)—$p_2 = 240$ kPa; (‑·‑·‑·‑)—$p_2 = 280$ kPa; (··‑··‑··)—$p_2 = 320$ kPa.

The static load from source foundation $p_1$ and the soil above the foundation base serve as the lateral surcharge concerning the nearby foundation. This lateral surcharge increases

mean stresses $\sigma_m$ and reduces the intensity of shear stresses $\tau_i$, thereby bringing the stress state a little closer to compression.

The above statement that the settlement of the nearby foundation $s_2$ decreases with increasing static pressure $p_1$ from the source foundation has some exceptions. As it can be seen, for large values of $p_1$ and small values of $p_2$ and $c$, the settlement of the nearby foundation $s_2$ increases with the growth of $p_1$, and does not decrease. The graph in Figure 19 shows the peculiar shape of the two characteristic areas (approximately $p_1 < 150$ kPa and $p_1 > 150$ kPa), which require explanation. The first area (approximately $p_1 < 150$ kPa) shows an intensive decrease in the $s_2$ settlement caused by the loading of the zone of the soil uplift from under the base of the adjacent foundation. In this case, the size of the zone of plastic deformation propagation increases because the intensity of tangential stresses $\tau_i$ decreases and, thus, gets further from their limiting value $\tau_i^*$ (Figure 17b). Since the ratio of stresses $\tau_i$ and $\tau_i^*$ has a non-linear character, in the area of $p_1 \approx 150$ kPa the effect of an increase in lateral loading decreases, and the curve changes its direction. At the same time, if the distance between the foundations $c$ is small and the load from nearby foundation $p_2$ is low, one can observe an increase in settlement $s_2$ in the case of considerable load from the source foundation $p_1$ (Figure 19, approximately $p_1 > 150$ kPa), which is explained by an increase in the intensity of shear stresses $\tau_i$ and small mean stress $\sigma_m$.

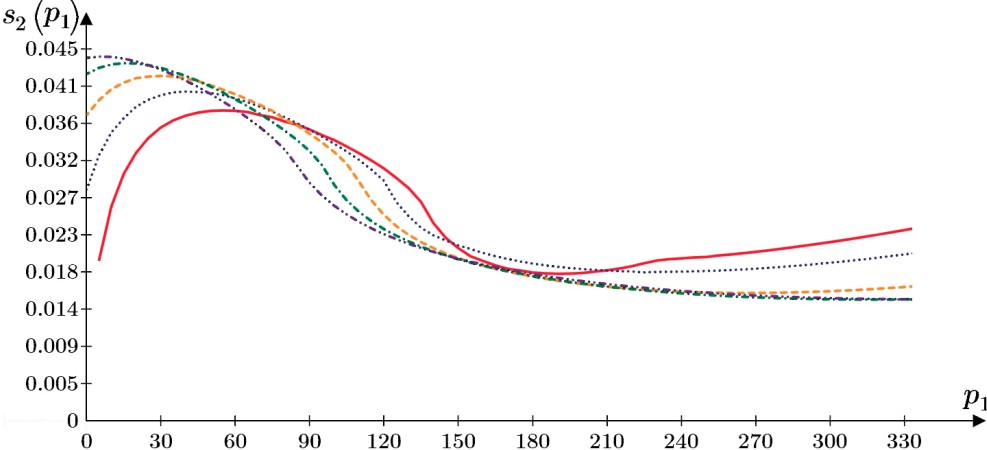

**Figure 19.** Dependence of the value of excessive settlement of the foundation of a nearby building $s_2$ on static load from the source foundation $p_1$, if load $p_2 = 90$ kPa, dynamic component $\Delta p = 30$ kPa, and the distance between the foundations $c$: (———)—$c = 0.0$ m; (··········)—$c = 0.5$ m; (-----)—$c = 1.0$ m; (–·–·–·–)—$c = 1.5$ m; (··–··–··)—$c = 2.0$ m.

The graph, showing the dependence of the value of excessive settlement of a nearby foundation $s_2$ on the static load from source foundation $p_2$, if the dynamic component $\Delta p = 30$ kPa and values of load $p_1$ are variable, is shown in Figure 20. Dependence of the effect of load from the source foundation $p_1$ on the value of excessive settlement of a nearby foundation $s_2$ can also be seen in Figure 16, where smaller values of load $p_1$ correspond to a larger value of settlement $s_2$.

If the static load from the source foundation $p_1 = 160$ kPa, the dynamic component is $\Delta p = 30$ kPa, the load $p_2 = 350$ kPa, and the values of depth $h_1$ vary, the dependence of the value of excessive foundation settlement $s_2$ on the distance between the foundations $c$ is as shown in Figure 21. The settlement of the nearby foundation $s_2$ nonlinearly decreases with an increase in the distance from the source foundation, which naturally follows from theoretical assumptions. At a distance of $c > 2b$, the effect of the transmitting foundation decreases significantly, and this fact converges with the numerical solution of Vivek P. and Ghosh P. [42] to the problem of the effect of a transmitting foundation that transmits static and dynamic loads to the base (an active foundation) on the additional settlement of an adjacent foundation that only transmits static loads to the base (a passive foundation).

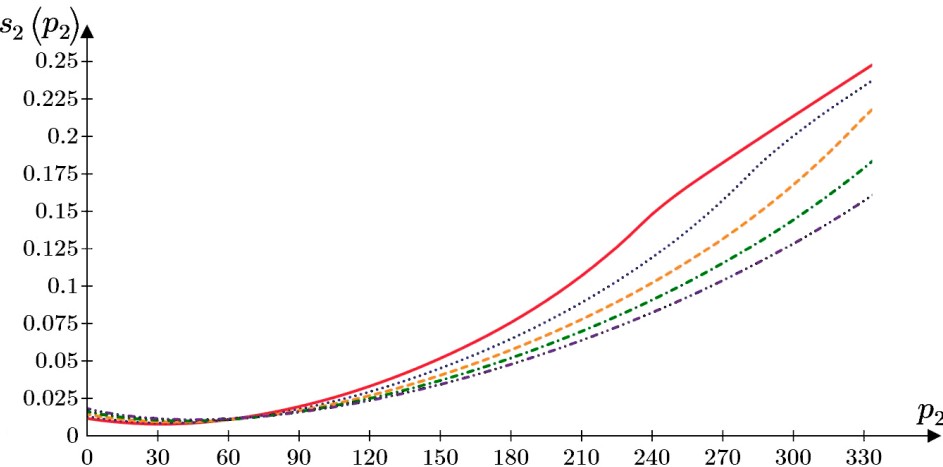

**Figure 20.** Dependence of the value of excessive settlement of a nearby building $s_2$ on static load from the source foundation $p_2$ with dynamic component $\Delta p = 30$ kPa and load $p_1$: (———)—$p_1 = 160$ kPa; (············)—$p_1 = 200$ kPa; (- - - - -)—$p_1 = 240$ kPa; (-·-·-·-)—$p_1 = 280$ kPa; (·-··-··-)—$p_1 = 320$ kPa.

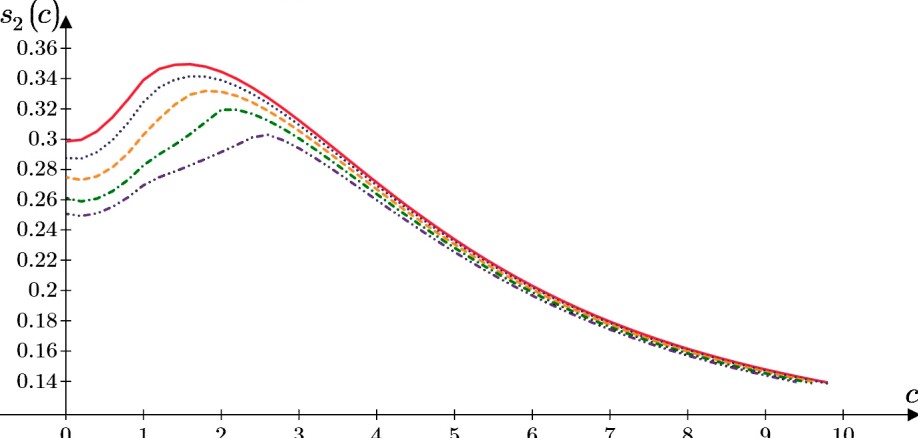

**Figure 21.** Dependence of excessive settlement of the foundation $s_2$ of a nearby building on the distance between foundations $c$, if the static load from the source foundation $p_1 = 160$ kPa, the dynamic component is $\Delta p = 30$ kPa, the load $p_2 = 350$ kPa, and the depth $h_1$: (———)—$h_1 = 1.0$ m; (············)—$h_1 = 2.0$ m; (- - - - -)—$h_1 = 3.0$ m; (-·-·-·-)—$h_1 = 4.0$ m; (·-··-··-)—$h_1 = 5.0$ m.

Settlement $s_2$ of the foundation located near the foundation that serves as the source of the dynamic effect (the source foundation) naturally increases with an increase in static pressure $p_2$ below its base (Figures 15 and 20); however, it decreases as the static component of the load from the source foundation $p_1$ and the depth of the foundation increase (Figures 16 and 19), which is explained by an increase in the difference between shear stress intensity $\tau_i$ and the limiting value $\tau_i^*$, the first of which decreases as the stress state of the base approaches its limit value and the second of which increases due to the increasing mean stress $\sigma_m$ (Figure 17). The theoretical solution to the problem, obtained by the current authors, complies with the results of the field studies conducted by V.S. Bogolyubchik, V.Y. Khain, and M.N. Goldstein [14–16] in respect of the effect of soil loading beyond the foundation on the vibrocreep intensity of a sandy base subjected to vibrations. It is quite natural that the excessive settlement of a nearby foundation decreases with increasing distance from the source foundation $c$ and depth of the foundation $h_1$ (Figure 21), which is explained by a decrease in the value of vibratory acceleration and an increase in the viscosity coefficient [5,6,25–28]. The problem is solved for the value of the loading frequency $\omega = 10$ Hz. With a change in vibration frequency, the qualitative

pattern of vibration will remain the same, but the settlement will take greater values, as was identified earlier [27,28,50].

The above-mentioned conclusions were obtained for the case of flexible load transfer to the base without taking into account the stiffness of the foundations themselves and the superstructures of the construction. In the future, the solution to this problem can be developed by including the stiffness of building structures in the structural analysis, which should lead to a decrease in additional settlement.

The authors believe that this research can be furthered by developing a generalized rheological model of sandy soil subjected to cyclic loading. In his earlier work, one of the authors (AZT-M) remarked that lateral pressure increases under dynamic loading [25], which indicates a change in Poisson's ratio. In addition, AZT-M found that the phenomenon of vibrocreep is more pronounced when shear deformation prevails [25]. Further, the authors believe that more research is needed to study the effect of the dynamic load intensity on Poisson's ratio. They also intend to create a quantitative description of the effect of the stress–strain state of sandy soil on the intensity of vibrocreep, taking into account the Nadai-Lode parameter [54], as was done by K. Wang et al. [33]. The above-listed further research undertakings will generate important practical results that will allow for a more accurate projection of displacement of the soil massif and pressure on substructures subjected to dynamic loading.

## 5. Conclusions

Applying the analytical solution proposed by the authors, it was found that the settlement of a single foundation to which vertical static and dynamic loads are transmitted increases with an increase in its static and dynamic components. The dynamic component, which is triggered by viscoplastic shear of the double-sided extrusion type, makes a greater contribution to settlement propagation. The increase in settlement accompanying an increase in the dynamic component is due to a decrease in the viscosity coefficient that accompanies an increase in the vibratory acceleration of the base, while the increase in settlement that accompanies an increase in the static component is caused by an increase in the intensity of shear stresses in the base and those stresses approaching their limiting values.

The settlement of a foundation located in proximity to the foundation that is the source of dynamic effects naturally increases with an increase in static pressure below its base but decreases with an increase in the static component of the load from the source foundation and greater foundation depth, which is explained by an increase in the difference between the values of shear stresses intensity and their limiting values, the first of which decreases as the stress state of the foundation gets closer to compression, and the second of which increases due to an increase in the mean stress.

**Author Contributions:** Conceptualization, methodology, software, formal analysis, investigation, resources, data curation, A.Z.T.-M.; writing—original draft preparation, A.Z.T.-M., A.N.S. and E.S.S.; writing—review and editing, A.Z.T.-M.; visualization, A.Z.T.-M., A.N.S. and E.S.S.; supervision, project administration, funding acquisition, A.Z.T.-M. All authors have read and agreed to the published version of the manuscript.

**Funding:** This research was supported by the Ministry of Science and Higher Education of the Russian Federation (project No. FSWG-2023-0004, «A system of territorial seismic protection of critical infrastructure facilities based on granular metamaterials with the properties of wide-range phonon crystals»).

**Data Availability Statement:** The data used to support the findings of this study are included within the article. The original details of the data presented in this study are available on request from the corresponding author.

**Conflicts of Interest:** The authors declare no conflict of interest.

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
