# Peer review of "Settlement of a Foundation on an Unsaturated Sandy Base Taking Vibrocreep into Account"

_axioms, doi:10.3390/axioms12060594_

Round 1

Reviewer 1 Report

The manuscript "Settlement of a foundation on an unsaturated sandy base with account for vibrocreep" deals with the determination of the settlement of a foundation subjected to static and dynamic loads.

The authors also analyze the settlement of foundation No. 2, located in the vicinity of foundation No. 1, subjected to dynamic loads.

Section 2 "Materials and methods" presents data on the foundation and soil, such as: width, foundation load, soil bulk density, angle of internal friction, Young's modulus and soil Poisson's ratio, load frequency, etc. foundation settlement in time S(t), using the principle of superposition, was presented as the sum of settlements due to static loads Sst (elastic settlement) and dynamic loads Sdyn (viscoelastic component). The elastic settlement Sst was determined by the "layer-by-layer summation of displacements" method. Settlement due to dynamic loads Sdyn was obtained by integrating over the depth of the layer of deformations εz, obtained by A.Z. Ter-Martirosyan and E.S. Sobolev. The strains εz are expressed by stresses, which were obtained by solving the Boussinesq problem in the case of 2-D (maybe it should be emphasized? It is only written that it is a flat problem, unless it is obvious).

In formula (2) there is a designation of stresses σzp, which are defined by formula (4). In formulas (4) we do not find the definition of these stresses σzp. Additionally, shouldn't formula (4) have 3 in the denominator instead of 3 (see [52], p. 64, formula 3.5)?

Figs. 3, 4 show the known contour diagrams of the average (hydrostatic) stresses and shear stresses, prepared for the characteristics of the foundation and subsoil considered in the work.

In the description of formula (7) it should be added, in my opinion, that the last part of the formula (after the second equal sign, i.e. b E(1+), according to [3] (p. 58, first paragraph), is recommended for adoption in the Soviet the standard for building design (based on S. Lapin, who tested more than 300 foundations with an area of 0.5 to 3700 m2) in the absence of experimental data In what units in formula (7) is the number 10?

The formula (9) lacks a description of the parameters and M (I guess it is the mass of the foundation). Also, formula (9) lacks the pcp description (I don't know if the cp index is not in Russian).

Fig. 13, in the "Results" chapter, shows the graphs of foundation settlement over time. In the description of Fig. 13, the authors write: "The analysis of Fig. 13 shows that with the increase of the static load component p1 and the dynamic load component Δp, the settlement of the foundation also increases." In my opinion, this conclusion is obvious.

In the section "Discussion" (lines 386-397) an explanation of the reduction of settlement with increasing load p1 is given, shown in Fig. 16. In my opinion, a graph of the stress difference τi and τi* should be provided against p1. It would also be necessary to add stress dependence diagrams τi and σm under the foundation No. 2 from the load p1 (the authors put more primitive graphs, e.g. 3,4,6, instead of really interesting ones).

The nature of the curves in Fig. 16 and Fig. 18 of the relationship between settlement S2 and load p1 is incomprehensible. The difference between these graphs is that Fig. 16 shows the relationship of settlement S2(p1) for different values of p2 (the distance between foundations c is not given), while Fig. 18 shows the relationship S2(p1) for p2 =90 kPa for different distances between foundations c (in Fig. 16, settlements disappear with increasing p1, and in Fig. 18 - they increase). In my opinion, the graphs are not reliable (or the explanatory information for these figures was incorrectly provided). It is incomprehensible why the authors assumed p2=90 kPa to prepare Fig. 18?

In Section 5 "Conclusions", the first sentence (belief 477-479) is very obvious and has long been known. The last paragraph of the conclusions (line 489) mentions the dependence of the settlement S2 on the value of the load p1 and the foundation depth. However, in the work we do not find studies on the impact of the foundation depth of foundation No. 1 on the settlement of S2.

The same conclusions regarding the description of graphs are often repeated in the work. In the literature, all the names of the works have been translated into English, despite the fact that these works are written in Russian (unless it is generally accepted). Such a translation of the name of works into English causes many problems with their dependence.

Almost half of the cited references are older than 10 years, although the most recent items from the last 2 years are also cited.

Reviewer 2 Report

This study is well written and organized in general. Hence, the reviewer has no further comments for it as a scientific paper. However, the limitation of this study that no experimental studies are carried out to validate the effectiveness should be mentioned in the conclusion.

Reviewer 3 Report

In the paper entitled “Settlement of a foundation on an unsaturated sandy base with account for vibrocreep”, an analytical solution to the problem of settlement of (1) the foundation that is the source of dynamic loading, and (2) the nearby foundation, taking into account the rheological properties of sandy soil subjected to vibration, given that these rheological properties depend on shear stresses. The proposed solution allows describing the progress of deformation over time.

COMMENTS

The revised paper is improved.

Some recommendations for future research were added at the end of the Conclusions.

Minor editing of English language required.

Round 2

Reviewer 1 Report

The authors made corrections as recommended by the reviews. The article is clearer. It is still possible to make minor corrections and additions to the description of the research methodology and results, as well as supplementing the literature with a few newer items. However, the article may be accepted for publication in its present form. Good luck!